# Therapeutic Targeting of Intestinal Fibrosis in Crohn’s Disease

**DOI:** 10.3390/cells11030429

**Published:** 2022-01-26

**Authors:** Giovanni Santacroce, Marco Vincenzo Lenti, Antonio Di Sabatino

**Affiliations:** First Department of Internal Medicine, San Matteo Hospital Foundation, University of Pavia, 27100 Pavia, Italy; giovanni.santacroce01@universitadipavia.it (G.S.); m.lenti@smatteo.pv.it (M.V.L.)

**Keywords:** antifibrotic therapy, Crohn’s disease, IBD, intestinal fibrosis, stricture

## Abstract

Intestinal fibrosis is one of the most threatening complications of Crohn’s disease. It occurs in more than a third of patients with this condition, is associated with increased morbidity and mortality, and surgery often represents the only available therapeutic option. The mechanisms underlying intestinal fibrosis are partly known. Studies conducted so far have shown a relevant pathogenetic role played by mesenchymal cells (especially myofibroblasts), cytokines (e.g., transforming growth factor-β), growth factors, microRNAs, intestinal microbiome, matrix stiffness, and mesenteric adipocytes. Further studies are still necessary to elucidate all the mechanisms involved in intestinal fibrosis, so that targeted therapies can be developed. Although several pre-clinical studies have been conducted so far, no anti-fibrotic therapy is yet available to prevent or reverse intestinal fibrosis. The aim of this review is to provide an overview of the main therapeutic targets currently identified and the most promising anti-fibrotic therapies, which may be available in the near future.

## 1. Introduction

Crohn’s disease (CD) is a chronic-relapsing immune-mediated disorder [1], with a prevalent gastrointestinal involvement and a constantly increasing incidence worldwide, especially in Western countries [2], representing a major concern for the healthcare system. The main symptoms experienced by CD patients include abdominal pain, diarrhea and fever, with a severe impairment of their quality of life [3,4]. One of the most common and threatening complications of CD is intestinal fibrosis, which occurs in more than a third of patients and leads to intestinal obstruction due to strictures [5]. Intestinal fibrosis results in increased morbidity and mortality, causing prolonged hospitalization and a need for surgery [6].

Fibrogenesis is a pathophysiological process through which one’s organism reacts to any type of damage due to noxious agents, such as physical, chemical and mechanical injury, infections, and autoimmunity [7]. The process of wound healing, which requires the intervention of a large number of molecular and cellular components, leads to the deposition of connective tissue in the extracellular matrix (ECM) in response to damage, resulting in tissue regeneration and repair [8]. However, when the stimulus to fibrogenesis becomes persistent or recurrent or even abnormal or exaggerated, as in the case of CD, this process may become uncontrolled [9], resulting in tissue fibrosis and scarring, with irreversible anatomical and/or functional alterations, eventually causing intestinal obstruction.

The mechanisms underlying gut fibrosis are only partly known and, although some pre-clinical studies have been conducted so far, currently there is no clinically feasible therapy to prevent or reverse fibrosis. The aim of this review is to provide a broad overview of currently known therapeutic targets and of most promising anti-fibrotic therapies that may shortly be available for clinical use.

## 2. Materials and Methods

In November 2021 we searched MEDLINE (PubMed) in a non-systematic manner by using the medical subject heading terms “fibrosis”, “strictures”, “intestinal fibrosis”, “anti-fibrotic therapy”, “target therapy” and “intestine”, “gut”, “inflammatory bowel disease”, and “Crohn’s disease” for all articles published since database inception. More than 270.000 papers were found, so we have restricted the search using the terms “fibrosis” and “Crohn’s disease”, finding 1.330 papers, most of which were either duplicates or non-original or not strictly related to the subject of this review. We selected only 145 studies (both non-human and human) exploring the mechanism of intestinal fibrosis (i.e., cellular, molecular, endoluminal, and molecular mechanism) in CD and the related therapeutic management, focusing on target therapy. We also searched the reference lists of pivotal review articles for additional papers that we judged to be relevant to this review. Figure 1 shows the flow-chart of our search strategy.

## 3. Overview of the Main Mechanisms of Intestinal Fibrosis in CD

As mentioned above, the abnormal inflammatory stimulus due to CD is associated with uncontrolled activation of mesenchymal cells, resulting in excessive ECM deposition [10]. In addition, an imbalance between matrix metalloproteinases (MMPs) and their inhibitors, tissue inhibitors of metalloproteinases (TIMPs), appears to be associated with increased ECM deposition and subsequent tissue fibrosis [11]. These mechanisms, together with the thickening of the muscle layer due to hyperplasia and hypertrophy of the smooth muscle cells [12], determine the development of fibrostenotic strictures in CD.

Here we will briefly outline the main players underlying the process of intestinal fibrosis, schematically represented in Figure 2. The correlation between these factors is in most cases unknown and elusive. Figure 3 shows some of these cellular and molecular players and their interaction in the fibrogenic process leading to intestinal stricture formation.

### 3.1. Main Cells Involved in Fibrogenesis

The main cells involved in the fibrogenesis process are mesenchymal cells, which are specifically committed to the production of collagen. The main players in this process are fibroblasts, myofibroblasts, and smooth muscle cells [13]. In particular, the inflammatory stimulus associated with CD seems to determine the activation of tissue fibroblasts and the migration of non-resident fibroblasts at the site of damage. These fibroblasts, under the stimulus of growth factors such as transforming growth factor (TGF)-β, may differentiate into myofibroblasts, capable of producing ECM. Similarly, smooth muscle cells are able to differentiate into myofibroblasts and likewise myofibroblasts can differentiate into smooth muscle cells and lead to the thickening of the muscularis propria and the formation of strictures [14]. Finally, the possible role of inflammation-induced differentiation of epithelial and endothelial cells into ECM-secreting mesenchymal cells should be considered, according to the mechanisms of epithelial-mesenchymal transition (EMT) and endothelial-mesenchymal transition (EndMT) [15,16]. EMT is a constantly evolving process in which epithelial cells acquire a migratory function and develop fibroblast characteristics. Similarly, EndMT is a process in which endothelial cells acquire fibroblast characteristics.

### 3.2. Molecular Mediators of Fibrosis

It is assumed that several cytokines can actively participate into the fibrogenesis process. Among these, the role of TGF-β is certainly predominant [17]. More specifically, the TGF-β1 isoform promotes collagen synthesis and fibroblast contraction in the mucosa of patients with fibrostenosing CD, acting through the Smad2-Smad3 molecular pathway and the regulation of TIMPs. Other cytokines related to organ fibrosis and with an emerging role in intestinal fibrosis, besides their known pro-inflammatory properties, are those belonging to the interleukin (IL)-1 family, including IL-1, IL-33, and IL-36 [18,19,20,21]. CD4+ T cells play a crucial role in the pathogenesis of CD and several T helper (Th) subsets have been identified, with different roles. While T-regulatory cells prevail in normal conditions, the Th1 subset appears predominantly pro-inflammatory, whereas Th2 and Th17 subsets appear to have both pro-inflammatory and pro-fibrogenic roles [22,23]. In particular, Th17 cells produce both IL-17 and IL-22 with a possible contrasting effect on intestinal fibrogenesis [24]. The role of cytokines belonging to the IL-17 family is well established, especially that of the IL-17A, as it induces intestinal myofibroblast secretion of collagen and TIMPs and significantly inhibits myofibroblast migration [25]. A possible role in this process has also been ascribed to the IL-17E (also known as IL-25), whose production in the human gut is reduced by tumor necrosis factor (TNF)-α and enhanced by TGF-β1 [26]. However, the pro-fibrotic role of IL-17E in CD has been questioned by the finding of no-significant difference on IL-17E levels in strictured compared to non-strictured CD tissues [25]. Fibroblast activation protein (FAP) is another protein typically produced by activated fibroblasts during wound healing and implicated in the fibrotic evolution of tissue damage [27]. FAP has been shown to be highly overexpressed in the submucosa and the muscle layer of stenotic CD, compared to non-stenotic CD [28]. In addition, other growth factors have an established role in gut fibrosis, especially the basic fibroblast growth factor (bFGF), which is overexpressed in patients with stricturing CD phenotype [29]. Concerning the role of TNF-α family members, there is growing evidence about the TNF-like cytokine 1A (TL1A), secreted from immune cells and binding the death domain receptor 3 (DR3) expressed on intestinal myofibroblasts [30]. TL1A is highly expressed in the fibrotic tissue of CD patients and a gene variant of the TL1A gene is associated with a higher risk of fibrotic strictures [31]. Finally, a possible role of neutrophil extracellular traps (NETs) has recently emerged. NETs are large, extracellular, web-like structures extruded by neutrophils under various conditions, especially immune response towards pathogens, representing a defense mechanism that, if dysregulated, can contribute to the pathogenesis of immune-related disorders [32]. NETs have been shown to mediate the in vitro activation of fibroblasts into myofibroblasts in fibrotic interstitial lung disease [33], and it has been suggested that this role may also be played in the gut [34].

### 3.3. MicroRNAs

MicroRNAs (miRNAs) are small non-coding ribonucleic acid (RNA) sequences that interfere with mRNA, causing, in most cases, an inhibition of translation [35]. The role of miRNAs on intestinal fibrosis in CD is relatively poorly established. Two families of miRNAs, miRNA-29 and miRNA-200, appear to be involved in this process. Specifically, miRNA-29a, -29b, and -29c were found to be down-regulated in CD strictured mucosa, with a role for miRNA-29b in modulating in vitro the expression of collagen I and III [36]. The miRNA 200 family appears to play a protective role against the development of EMT [37,38].

### 3.4. The Role of Gut Microbiota

The human gut hosts a complex and abundant aggregation of microbes, collectively referred to as the gut microbiota, whose compositional and metabolic alterations, defined as dysbiosis, have a pivotal role in IBD pathogenesis [39]. All intestinal immune and non-immune cell types express the pathogen recognition receptors, such us Toll-like receptors (TLRs) and nucleotide-binding and oligomerization domain (NOD)-like receptors, which provide the ability to respond to pathogen-associated molecular patterns [40]. It has been reported that in primary human intestinal fibroblasts, the TLR5 ligand flagellin (present in all flagellated bacteria) induces a pro-inflammatory and pro-fibrotic phenotype [41]. More recently, it was confirmed that flagellin derived from adherent-invasive Escherichia coli (AIEC), a micro-organism frequently isolated in the ileal tissue of CD, could bind the TLR5 expressed in intestinal epithelium, determining the expression of the IL-33 receptor (ST2), which is crucial for the development of intestinal fibrosis [42]. Supporting the hypothesis of a causal role of microbiota on intestinal fibrogenesis, the recurrence of NOD2 variants in CD patients with fibrostenotic phenotype has been highlighted [43]. Finally, several studies on animal models of intestinal fibrosis have shown a pro-fibrotic activity of the gut microbiota [44,45].

### 3.5. Matrix Stiffness

Another novel mechanism associated with intestinal fibrosis in CD patients is matrix stiffness. Resistance to matrix deformation has been shown to be an important mediator of cellular behavior [46]. Cell proliferation and differentiation are assumed to increase with matrix stiffness. This is true even for human colonic fibroblasts, which have been proved to be activated to a profibrotic phenotype by matrix stiffness [47,48]. An increase in ECM stiffness would be associated with a morphological alteration of fibroblasts, actin stress fiber formation and focal adhesion, promoting fibroblast proliferation and activation, even in the absence of inflammatory stimulation. This finding suggests that intestinal fibrosis would have a self-propagation mechanism independent of the inflammatory stimulus and matrix stiffness may have a role in this process.

### 3.6. “Creeping Fat”

Mesenteric fat and its hypertrophy, known as ‘creeping fat’, have an emerging role in the pathogenesis of CD fibrostenotic phenotype [49]. The role of creeping fat in intestinal fibrogenesis is associated with the production of adipokines that promote a shift of the macrophage compartment towards M2 macrophage dominance, resulting in increased production of TGF-β1 [50]. Furthermore, creeping fat has been shown to be associated with smooth muscle hypertrophy [51].

## 4. Diagnostic Tools 

The diagnosis of intestinal fibrosis is usually established when the strictures become clinically manifest. With regard to diagnostic tools, biomarkers (such as fecal calprotectin and C-reactive protein), non-invasive imaging techniques (barium contrast studies and cross-sectional imaging), and invasive techniques (endoscopy and histology) are worth mentioning. An expert consensus about the definition and diagnosis of strictures in CD has recently been published, showing that cross-sectional imaging or ileocolonoscopy alone are appropriate to diagnose a small bowel stricture [6]. Magnetic resonance (MR) enterography is, at present, considered the best imaging technique. Localized luminal narrowing (luminal diameter reduction of at least 50%), bowel wall thickening (25% increase in wall thickness relative to the adjacent non-affected bowel), and pre-stricture dilatation (luminal diameter greater than 3 cm) on cross-sectional imaging are considered the most accurate radiological findings for stricture definition.

A relevant diagnostic issue is the distinction between an inflamed or a fibrotic stricture, which is crucial for treatment: predominantly inflammatory strictures may benefit from medical anti-inflammatory treatment while predominantly fibrotic strictures need endoscopic or surgical approach [52]. Unfortunately, there is often an overlap between inflammatory and fibrotic components and currently we do not have a technique able to accurately distinguish between these two components, since both histology and cross-sectional techniques have not shown sufficient accuracy.

Another challenging issue is the diagnosis of stricture recurrence in the post-operative setting. The identification of patients undergoing early symptomatic CD recurrence could be useful for a timely treatment and possible relapse prevention [53]. Ileocolonoscopy represents the gold standard for assessing CD recurrence, but more recently a non-invasive procedure, namely small intestine contrast ultrasonography (SICUS), has been proposed as a valid alternative. SICUS, which provides an extraluminal surface visualization, has in experienced hands shown comparable findings to endoscopy. Thus, its use may be useful for proper follow-up and treatment of patients after ileo-colonic resection for CD. 

## 5. Therapeutic Approaches

To date, there are no medical therapies available to prevent or reverse intestinal fibrosis in CD. In patients with intestinal obstruction due to CD with a fibrostenosing phenotype, initial treatment consists of nasogastric decompression, bowel rest, intravenous hydration, and electrolyte replacement. Subsequent management is dependent on strictures’ inflammation degree and morphometrics, such as location and length, assessed by biomarkers (e.g., C-reactive protein, erythrocyte sedimentation rate, and fecal calprotectin), endoscopy, computed tomography, or magnetic resonance imaging. Current complications (such as phlegmon, abscess, dysplasia, or malignancy) and patient preferences should also be taken into account [5,54].

### 5.1. Current Medical Options

Strictures with a dominant inflammatory component may benefit from anti-inflammatory therapy, which could reduce wall edema and thus intestinal wall thickness. Data on corticosteroids are still limited and controversial. Although these drugs seem to provide short term relief of symptoms, they do not decrease the need for surgery over time, nor do they improve patient prognosis [55]. Moreover, chronic steroid use may increase the risk of developing fibrosis and scarring. Steroid-dependent or refractory patients may benefit from biological drugs. Currently, there are only few data on anti-TNF agents. At first, some evidence led to concerns about this approach, given the apparent worsening of stenosis due to anti-TNF, caused by the rapid healing of ulcers [56,57]. However, more recent data have overcome this old assumption, demonstrating the efficacy of anti-TNF in stricturing CD in up to about two-thirds of the patients [58,59]. In particular, a prospective multicentric observational cohort study (CREOLE) tested the effect of induction and maintenance therapy with adalimumab in patients with stenosing CD, demonstrating its safety and efficacy in maintaining surgery-free remission in half of the included patients over a 4-year follow-up [60]. However, the surgery-free interval is still short and data on a direct anti-fibrotic effect of anti-TNF therapy are still lacking. With regard to other biologic drugs, vedolizumab (anti-integrin α4β7) has been shown to be effective and safe in real-world studies even in stricturing CD [61], whilst a recent multicentre study showed that ustekinumab (anti-IL12/23) is associated with a lower likelihood to achieve remission at six months in these patients [62]. As for 5-aminosalicylic acid (5-ASA), it does not appear to have anti-fibrotic activity and fails to medically induce remission in CD [63]. There is currently no evidence that purine analogues may be effective for symptomatic stricturing CD.

### 5.2. Endoscopic and Surgical Management

If anti-inflammatory therapy is not effective in relieving obstructive symptoms or if symptoms recur within a short period, endoscopic therapy, strictureplasty, or bowel resection should be considered [5]. As mentioned above, the decision between these choices must be taken with a multidisciplinary approach, taking into account the features of the strictures, the accompanying complications, the length of symptom-free interval, and the patient preference [64].

With regards to the endoscopic approach, endoscopic balloon dilatation (EBD) is considered the preferred technique for selected CD strictures, as it has proven a high rate of short-term technical and clinical efficacy, with substantial long-term efficacy and acceptable rates of complication [65,66]. EBD is a minimal-invasive procedure that consists in placing a radial expanding balloon dilator (available in an array of designs, lengths, and calibers) in the stenotic tract and inflating the balloon as needed [67]. Experts have judged the following items relevant in practicing EBD: 18 mm as the maximal luminal diameter after dilatation in one or several sessions, a balloon inflation time of at least 1 min, and 5 cm as the maximum stricture length that should be dilatated [6]. Furthermore, such a procedure has been defined successful when it is possible to pass an adult ileocolonoscope through a previously non-traversable stricture with a reasonable amount of pressure applied. 

When medical or endoscopic therapy fails or is contraindicated, surgery should be considered. According to an expert consensus, in stricturing disease both surgical resection and strictureplasty are valid options, with similar safety, efficacy, and long-term recurrence rates [68]. The preferred treatment of multiple fibrotic strictures of the small intestine, when technically feasible, are stricureplasties. According to the length and site of the stricture, multiple techniques for strictureplasty have been proposed, including Heineke-Mikulicz, Michelassi, and the Finney technique. In order to reduce fibrosis recurrence, multiple surgical strategies have been explored, including special anastomotic configurations, such as the antimesenteric functional end-to-end handsewn anastomosis, also known as Kono-S anastomosis [69], mesentery and lymph node excision [70], and the laparoscopic approach [71].

### 5.3. Promising Anti-Fibrotic Therapy in CD

Increasing knowledge of the molecular mechanisms underlying intestinal fibrosis has enabled the identification of anti-fibrotic therapeutic targets. At present, although there is no therapy capable of treating or reversing intestinal fibrosis in CD, several pre-clinical studies have been conducted in vivo, ex vivo, and in vitro, with encouraging results. Herein, we report the most promising anti-fibrotic therapeutic targets known to date and the relevant target-specific molecules under investigation (Table 1).

#### 5.3.1. Targeting TGF-β Pathways

The most promising target for anti-fibrotic therapy is TGF-β, the principal molecular mediator of fibrogenesis, and its signaling pathways.

Several studies on fibrosis of other tissues have shown that TGF-β1 production was strongly stimulated by the local activation of angiotensin II [72,73,74], the main effector of the renin-angiotensin system, whose activity is increased in the colonic mucosa of CD patients [75]. For this reason, it was assumed that angiotensin conversing enzyme (ACE) inhibitors and sartans (angiotensin II receptor antagonists), which typically act as anti-hypertensives, could also play a role in the process of intestinal fibrogenesis. The first ACE-inhibitor investigated was captopril, which showed to be effective in preventing colonic fibrosis in 2,4,6-trinitrobenzene sulfonic acid (TNBS)-induced colitis in rats. Its anti-fibrotic action has been assumed to derive from blocking TGF-β1 overexpression and/or from a direct down-regulation of TGF-β1 transcripts [76]. Moreover, transanal administration of enalaprilat has been shown to be effective in preventing colonic fibrosis in a dextran sulfate sodium (DSS)-induced colitis model [77]. More recently, losartan, an antagonist of the angiotensin II receptor, was investigated and exhibited a pleiotropic effect, reducing TGF-β1 concentration and significantly improving the macro- and microscopic scores of fibrosis in the colonic wall of rats [78];Based on the known antagonistic relationship between the TGF-β/Smad pathway and the peroxisome proliferator-activated receptor (PPAR)γ, a member of ligand-activated transcription factors of nuclear hormone receptor superfamily [79,80], the effect of a novel 5-ASA analog (named GED-0507-34 Levo), able to activate PPARγ, has been investigated. GED-0507-34 Levo showed improvement of intestinal fibrosis in DSS-induced chronic colitis in mice, reducing the activation of myofibroblasts and the expression of the main pro-fibrotic molecules including TGF-β, Smad3, IL-13 and connective tissue growth factor (CTGF) [81]. Similarly, it has been shown that other PPARγ agonists, usually employed in the treatment of diabetes, such as troglitazone and rosiglitazone, may be useful in counteracting the fibrogenic process by suppressing TGF-β1-induced synthesis of collagen, fibronectin, and α-smooth muscle actin in human primary intestinal myofibroblasts [82];Another target signaling pathway induced by TGF-β1 but also by matrix stiffness is that of Rho/Rho chinase (ROCK) [83]. The first ROCK inhibitors studied were CCG-1423, CCG-100602, and CCG-203971, which, by inhibiting RhoA signaling in myofibroblasts, induced a significant anti-fibrotic activity [84,85]. These molecules, however, showed an unacceptable toxicity profile, especially with regard to cardiovascular side effects [86]. For this reason, the effect of a locally acting ROCK inhibitor (AMA0825) was investigated. This molecule prevented and reversed intestinal fibrosis in vitro and ex vivo by diminishing TGF-β1-induced activation of myocardine-related transcription factor and p38 mitogen-activated protein kinase (MAPK) and increasing autophagy in fibroblasts, with a good tolerability profile [87]. Combining AMA0825 with anti-inflammatory agents (such as anti-TNF-α) in vivo ameliorated inflammation but also prevented accumulation of fibrotic tissue, underscoring the importance of combination therapy;Other compounds have been shown to downregulate the TGF-β signaling. These include cilengitide, which is an Arg-Gly-Asp (RGD)-containing αVβ3 integrin inhibitor, that is able to decrease TGF-β1 activation and development of fibrosis in chronic TNBS-induced colitis [88]. More recently, anti-fibrotic intestinal efficacy has been proposed for two molecules approved for the treatment of idiopathic pulmonary fibrosis, namely pirfenidone and nintedanib [89,90]. In particular, pirfenidone, an orally delivered pyridine derivative that suppresses TGF-β and TNF-α signals, inhibited, both in vivo and in vitro, intestinal fibroblast proliferation and motility and reduced collagen production through different TGF-β1 signaling pathways, including those of suppressor of mothers against decapentaplegic (Smad), phosphatidylinositol-3-kinase (PI3K)/AKT, MAPK, and mechanistic target of rapamycin (mTOR) [91,92,93,94]. Therefore, this molecule is of great interest and has important therapeutic potential, but needs further studies to better clarify its mechanism of action, efficacy, and safety [95]. No studies are yet available on the usefulness in intestinal fibrosis of nintedanib, a small oral molecule inhibitor of tyrosine kinase receptors, such as platelet-derived growth factor (PDGF), fibroblast growth factor (FGF), and vascular endothelial growth factor (VEGF) receptors. Finally, an anti-fibrotic action of maggot extract was described by downregulating the TGF-β1/Smad pathway via upregulation of nuclear factor erythroid 2-related factor 2 (Nrf2) expression [96].

#### 5.3.2. Targeting TIMP/MMP Balance

Intestinal fibrosis in CD is mainly due to the imbalance of deposition and degradation of ECM, regulated also by MMPs and TIMPs [11]. Thalidomide, a molecule with anti-inflammatory activity and emerging as an alternative treatment for refractory CD [97], has been shown to inhibit in vivo intestinal fibrosis by regulating TIMP/MMP protein balance and degradation of ECM [98].

#### 5.3.3. Targeting VEGF

The deposition of collagen causes chronic hypoxia, which in turn stimulates neo-angiogenesis through the upregulation of VEGF, thus favoring the deposition of further fibrotic tissue in a vicious circle [29]. VEGF has been supposed to be a therapeutic target of fibrosis, and its blockade through a monoclonal antibody (bevacizumab) has been investigated as a possible anti-fibrogenic strategy. However, this molecule showed a significant increase in fibrosis-related inflammatory cytokines in vitro [99] and, due to the possible side effects, could even worsen CD [100]. For this reason, targeting neo-angiogenesis does not currently seem to be useful, but rather harmful. Studies with anti-VEGF agents on models of intestinal fibrosis are still lacking.

#### 5.3.4. Targeting FAP

The FAP protein, discussed previously, could be a unique therapeutic target as it is a marker of active fibroblasts [101]. Thus, treatment directed against FAP would have high specificity and minimal side effects. Ex vivo treatment of stenotic tissues with anti-FAP antibody induced a dose-dependent decrease in collagen, particularly type I collagen, and TIMP-1 production, without altering MMP-3 and MMP-12 secretion [28]. Another FAP inhibitor to be mentioned is talabostat mesilate (PT100), which in a murine model of pulmonary fibrosis showed an anti-fibro-proliferative effect [102], but has not yet been studied for intestinal fibrosis.

#### 5.3.5. Targeting EMT

Another therapeutic target of recent interest is EMT, the complex process in which epithelial cells lose their phenotypic and functional characteristics and develop mesenchymal features [15]. In addition, it would appear that EMT may associate with intestinal fibrosis not only through direct production of myofibroblasts, but also through the release of crucial signals for myofibroblast differentiation [103,104]. Recent data from animal models of renal, hepatic, and cardiac fibrosis have demonstrated the anti-fibrotic effect of recombinant human bone morphogenic protein-7 (rhBMP-7) [105,106,107]. BMP-7 is a member of the TGF-β superfamily with the ability to counteract the pro-fibrotic action of TGF-β1. In vitro and in vivo studies have shown the effect of rhBMP-7 in inhibiting TGF-β1 induced EMT associated with intestinal fibrosis [108]. In addition, a recent study showed that miRNA200b-containing microvesicles inhibited colonic fibrosis, thus suppressing the development of EMT by targeting zinc finger E-box binding homeobox (ZEB)1 and ZEB2 [109].

#### 5.3.6. Targeting the Endogenous Cannabinoid System

The cannabinoid system comprises specific G-protein-coupled receptors (CB1 and CB2), a variety of exogenous (marijuana-derived cannabinoids) and endogenous ligands, and a machinery dedicated to endocannabinoid synthesis and degradation [110]. One of the main endogenous CB1 and CB2 agonists is anandamide (AEA) [111]. Given the evidence from experimental studies that the endocannabinoid system is involved in intestinal diseases and played a role in antagonizing fibrosis in chronic liver disease [112,113], the in vitro effect of the AEA analogue methanandamide (MAEA) on CD strictured myofibroblasts was investigated [114]. The CB2 agonist showed to reduce collagen production by strictured CD myofibroblasts and increase their migration ability. No further data are available about the anti-fibrotic role of cannabinoid receptor agonists.

#### 5.3.7. Targeting IL-17

As mentioned above, the IL-17A is overexpressed in CD strictures and determines myofibroblasts production of collagen and TIMP-1 and reduction of their migratory ability [25]. However, IL-17 contribution to IBD is still controversial [115]. A recent study has demonstrated that treatment with the anti-IL17 antibody, in TNBS-induced intestinal fibrosis mice, not only significantly decreased profibrogenic cytokines (IL-1β, TGF-β1, and TNF-α) and intestinal inflammation, but also reduced fibrogenesis-related TIMP-1 and MMP-2 gene expression [116]. Another recent study demonstrated a similar function of IL-17 in intestinal fibrosis, showing that IL-17-driven fibrosis is inhibited by Itch-mediated ubiquitination of hydrogen peroxide-inducible clone 5 (HIC-5) [117]. However, previous clinical trials reported a contradictory effect of anti-IL-17 treatment in CD patients, as blocking IL-17 with specific antibodies (secukinumab and brodalumab) failed to relieve symptoms and even increased disease activity in active CD patients [118,119]. The reason for this conflict could lie in the unclear role of IL-17 in the intestine immune homeostasis [24]. Thus, more investigations on the effect of anti-IL-17 treatment in intestinal fibrosis and on the safety of this therapy are needed.

#### 5.3.8. Targeting IL-36

IL-36 is a member of the IL-1 superfamily and consists of three agonists and one receptor antagonist (IL-36Ra) [21]. Endogenous agonists act as proinflammatory cytokines and the IL-36 signaling also promotes secretion of pro-fibrotic mediators. Thus, a potential role of IL-36R inhibition as a therapeutic strategy to treat pro-fibrotic disorders has been proposed [120]. Antibodies against IL-36R were investigated in DSS or TNBS-induced mice colitis and showed to significantly reduce established fibrosis [121]. Further studies are needed to ascertain the therapeutic potential of IL-36R signaling modulation in CD patients. A phase 2 trial is currently under way to evaluate the spesolimab (an anti-IL-36 receptor antibody) efficacy in patients with moderate-to-severe ulcerative colitis (NCT03482635).

#### 5.3.9. Targeting TL1A

Accumulating evidence demonstrated the importance of TL1A in the pathogenesis of IBD and suggested a potential therapeutic role of TL1A blocking [30,31]. More recently, anti-TL1A antibody injection showed to ameliorate intestinal fibrosis by inhibiting the activation of intestinal fibroblasts and reducing collagen deposition in the T cell transfer model of chronic colitis in mice [31]. This effect may be related to the inhibition of TGF-1/Smad3 signaling pathway. A Phase 2a, multicenter, single-arm, open-label study demonstrated an acceptable safety profile for the anti-TL1A antibody (PF-06480605), which was effective in inducing endoscopic improvement in adults with moderate-to-severe ulcerative colitis [122]. A phase 2b trial with this drug is in progress (NCT04090411).

#### 5.3.10. Targeting Both TNF-α and IL-17

ABT-122 is a novel bispecific dual variable domain immunoglobulin targeting human TNF-α and IL-17. It has been demonstrated to be safe and effective in rheumatoid arthritis and psoriatic arthritis [123,124]. The use of this molecule in immune-mediated intestinal diseases has recently been hypothesized, but no studies have yet been performed [125].

#### 5.3.11. Targeting AXL Pathway

AXL is a receptor tyrosine kinase that has been implicated in fibrogenic pathways involving myofibroblast activation [126]. A recent study demonstrated a role of AXL pathway in models of intestinal fibrosis and suggested that the inhibition of AXL signaling through small molecule inhibitor (BGB324) could represent a novel target to antifibrotic therapy for intestinal fibrosis, inhibiting both matrix-stiffness and TGF-β1-induced fibrogenesis in human colonic myofibroblast [127]. In addition, AXL inhibition sensitized myofibroblasts to undergo apoptosis.

#### 5.3.12. Targeting NETs

The potential role of NETs in intestinal fibrosis has already been mentioned, although data in the literature are still scarce and sometimes conflicting. In some studies, CD inflamed ileum has shown high expression of NETs [128,129], whereas in others no significant amount of NETs has been shown in CD when compared to ulcerative colitis [130,131]. The key process promoting NET formation is H3-citrullination-mediated by peptidylarginine deiminase 4 (PAD4), and studies in mice on pulmonary fibrosis have shown a reduction in fibrosis by suppression of PAD4 and consequently of NETs [132]. It has been suggested that PAD4/NETs inhibition may have a therapeutic role in CD as well [34], however no studies have yet been performed.

#### 5.3.13. Targeting miRNAs

miRNAs are increasingly studied as potential targets of anti-fibrotic therapies but no drugs targeting miRNAs are currently available in clinical practice. As said before, there is a significant down-regulation of the miRNA-29 family in the mucosa of CD strictured gut and it has been observed that the TGF-β1-induced collagen expression is reversed by exogenous overexpression of miRNA29b [36]. In addition, the administration of miRNA200 has shown to partially protect intestinal epithelial cells from fibrogenesis in vitro, through the repression of ZEB1 and ZEB2 and the supposed inhibition of EMT [38]. In the near future, miRNA modulation may provide interesting new therapeutic options.

#### 5.3.14. Targeting Matrix Stiffness

Modifications of physical environment can affect myofibroblast behavior and survival [41]. In vitro experiments showed that upon culture in a fibrotic environment, normal myofibroblasts increased the expression of MMPs, to counteract the mechanical force exerted by the matrix, by expressing increased levels of the collagen crosslinking enzyme lysyl oxidase (LOX), and inducing more ECM contraction [133]. LOX inhibition completely restored MMP3-activity in CD stenotic myofibroblasts and prevented ECM contraction, allowing to consider LOX a potential anti-fibrotic agent.

#### 5.3.15. Targeting Intestinal Microbiota

Given the increasing emphasis on the pro-fibrotic impact of the gut microbiota, many in vitro and in vivo studies have been carried out to assess the effect of probiotics and prebiotics on intestinal fibrosis [134,135,136,137]. Among these, the most recent suggested that the soluble fraction of Vivomixx^®^ formulation was able to inhibit collagen-I and α-SMA expression in human colonic fibroblast by interfering TGF-β1/Smad2-3 signaling [138]. All the data available are still preliminary and need to be confirmed and expanded.

**Table 1 cells-11-00429-t001:** Therapeutic targets studied for intestinal fibrosis in Crohn’s disease.

TARGET	AGENT	MECHANISM	MODEL	REFERENCE
**TGF-β** **pathways**	Captopril	↓ *TGF*-β*1* expression and/or *TGF*-β*1* transcript	TNBS-colitis	[76]
	Transanal enalaprilat	↓ *TGF-*β signaling pathway	DSS-colitis	[77]
	Losartan	↓ *TGF*-β*1* expression	TNBS-colitis	[78]
	GED-0507-34 Levo	PPAR-γ activation	DSS-colitis	[81]
	Troglitazone, Rosiglitazone	PPAR-γ activation	HIFs	[82]
	CCG-1423, CCG-100602, CCG-203971	ROCK inhibition	CCD18-co HIFs	[85]
	AMA0825	ROCK inhibition	DSS- and T-cell transfer-colitis, HIFs	[87]
	Cilengitide	αVβ3 integrin inhibition	TNBS-colitis	[88]
	Pirfenidone	Smad, PI3K/AKT, MAPK, and mTOR signaling pathways inhibition	HIFs, DSS-colitis, RIF	[91,92,93,94]
	Maggot extract	↑ Nrf2 expression	DSS-colitis	[96]
**TIMP/MMP balance**	Thalidomide	Altered TIMP/MMPs balance and ECM degradation	TNBS-colitis	[98]
**VEGF**	Bevacizumab	↓ collagen deposition	n.a.	n.a.
**FAP**	Anti-FAP Ab	FAP inhibition	HIFs	[28]
**EMT**	rhBMP-7	EMT inhibition	TNBS-colitis	[108]
	miRNA200b-containing microvescicles	EMT inhibition	TNBS-colitis, IEC-6	[109]
**Endogenous cannabinoid system**	MAEA	↓ collagen production and ↑ myofibroblasts migration	Human organ culture biopsies, LPMCs, and HIFs	[114]
**IL-17**	Anti-IL17 Ab	↓ profibrogenic cytokines and MMP/TIMPs balance alteration	TNBS-colitis	[116]
**IL-36**	Anti-IL36R Ab	↓ collagen production, MMPs, IL6 signaling, and EMT	DSS- and TNBS-colitis	[121]
**TL1A**	Anti-TL1A Ab	TGF-1/Smad3 signaling pathway inhibition	T-cell transfer-colitis	[31]
**TNF-** **α** **and** **IL-17**	ABT-122	n.a.	n.a.	n.a.
**AXL pathway**	BGB324	↓ matrix stiffness and TGF-β1-induced fibrogenesis	CCD-18co, TNBS-colitis	[127]
**NETs**	PAD4 inhibitors	↓ NETs-derived fibrosis	n.a.	n.a.
**miRNA**	miRNA29	↓ TGF-β1-induced collagen expression	Human fibroblasts cultures	[36]
	miRNA200	↓ ZEB1 and ZEB2, EMT inhibition	Intestinal epithelial cells	[38]
**Matrix Stiffness**	β-aminopropionitrile	↑ MMP3 activity and ↓ ECM contraction	HIFs	[133]
**Gut microbiota**	Probiotics and prebiotics	Modulation fibrotic pathways	Mouse and cellular models	[134,135,136,137,138]

Abbreviations: Ab, antibody; CCD-18Co, noncancerous colon fibroblast; DSS, dextran sulfate sodium; ECM, extracellular matrix; EMT, epithelial-mesenchymal transition; EndMT, endothelial mesenchymal transition; FAP, fibroblast activation protein; HIF, human intestinal fibroblast; IEC, intestinal epithelial cell; IL, interleukin; LPMC, lamina propria mononuclear cell; MAEA, methanandamide; MAPK, mitogen-activated protein kinase; mTOR, mechanistic target of rapamycin; miRNA, micro ribonucleic acid; MMP, matrix metalloproteinase; n.a., not available; NET, neutrophil extracellular trap; Nrf2, nuclear factor erythroid 2-related factor 2; PAD4, peptidylarginine deiminase 4; PI3K, phosphatidylinositol-3-Kinase; PPAR, peroxisome proliferator-activated receptor; rhBMP-7, recombinant human bone morphogenic protein-7; RIF, radiation-induced intestinal fibrosis; ROCK, Rho/Rho chinase; Smad, suppressor of mothers against decapentaplegic; TGF, transforming growth factor; TIMP, tissue inhibitor of metalloproteinase; TL1A, TNF-like cytokine 1A; TNBS, 2,4,6-trinitrobenzene sulfonic acid; TNF, tumor necrosis factor; VEGF, vascular endothelial growth factor; ZEB, zinc finger E-box binding homeobox; ↑, increase; ↓, decrease.

## 6. Major Challenges for Anti-Fibrotic Agents Development

There are numerous pitfalls in identifying anti-fibrotic drugs for CD. First, experimental fibrosis in cellular and animal models does not necessarily resemble human fibrosis [139]. Cells may behave differently in vitro and in vivo and single cell studies often do not reproduce the complex in vivo cellular network. For this reason, 3D models are under development to reproduce the natural microenvironment as closely as possible to that in vivo [140,141]. Moreover, the targeted molecules often represent a small component of the complex molecular maze underlying the fibrotic process. The target molecules may even have multiple functions on the intestinal tissue, with the risk of targeting processes implicated in physiological tissue remodeling, resulting in negative effects. In addition, to date there are neither fibrosis biomarkers, nor diagnostic tools that can be used to identify and quantify the overall fibrotic burden in CD patients, especially in the early stages, when anti-fibrotic therapy may be mostly effective [142,143]. Finally, there is an urgent need of end points that can be used to assess the efficacy of anti-fibrotic agents in clinical trials. For this reason, several groups of renowned IBD experts have reached expert consensus on this matter [6,144]. In particular, a core set of 13 end-points (i.e., complete clinical response, long-term efficacy, sustained clinical benefit, treatment failure, radiological remission, normal quality of life, clinical remission without steroids, therapeutic failure, deep remission, complete absence of occlusive symptoms, symptom-free survival, bowel damage progression, and no disability) were considered critical [144]. The combination of improved clinical, endoscopic and/or radiological features seems appropriate to define a successful treatment [6]. The need for intervention within 24-48 weeks from medical therapy has been proposed as the most accurate end-point to assess anti-fibrotic agents in pharmacological trials.

Despite the urgency for anti-fibrotic therapy and the numerous molecules identified as potential anti-fibrotics in CD, no phase III clinical trial is currently ongoing or recruiting to our knowledge (according to ClinicalTrials.gov, as of 19 January 2022).

## 7. Conclusions

Nearly half of all patients with CD will develop intestinal strictures along the disease natural history. At present, despite substantial progresses in our understanding of the pathophysiological mechanisms underlying fibrosis, there remains a translational gap between the identification of putative anti-fibrotic targets and translation into effective therapies in humans [145]. Endoscopic and surgical approaches are currently the only options available and there is an urgent need for targeted anti-fibrotic therapy. Several molecules investigated in preclinical studies, which are awaiting clinical trials in humans, have proven effective in CD stricturing phenotype and may be available in the near future as additional weapons in preventing or reversing intestinal fibrosis. The development of experimental models that will be increasingly overlapping with human fibrosis, the identification of serum biomarkers and diagnostic tools that can identify and predict the evolution of fibrostenosing disease, and the finding of endpoints suitable for clinical trials, represent major challenges for the development of clinically available anti-fibrotic agents for CD strictures.

## Figures and Tables

**Figure 1 cells-11-00429-f001:**
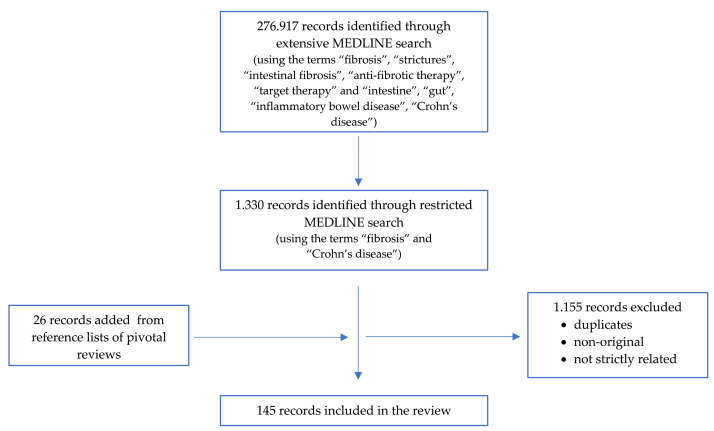
Flow-chart of the search strategy.

**Figure 2 cells-11-00429-f002:**
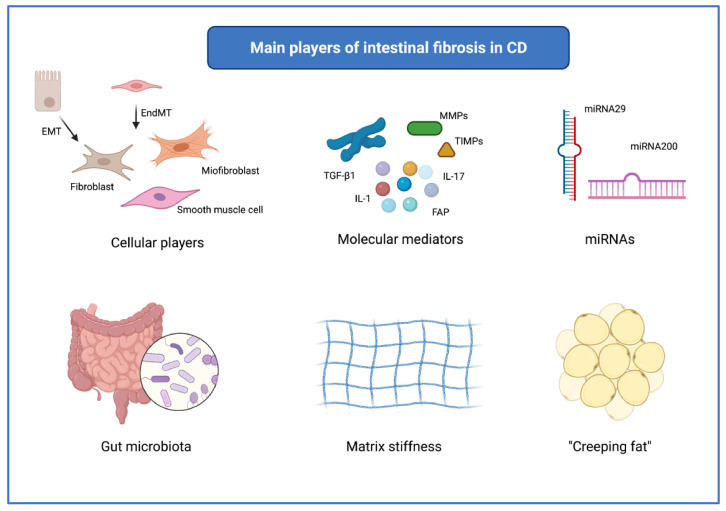
Schematic representation of the main players of intestinal fibrosis in Crohn’s disease. Abbreviations: CD, Crohn’s disease; EMT, epithelial-mesenchymal transition; EndMT, endothelial-mesenchymal transition; FAP, fibroblast activation protein; IL, interleukin; miRNA, micro ribonucleic acid; MMP, matrix metalloproteinase; TGF, transforming growth factor; TIMP, tissue inhibitor of metalloproteinase. Created with “BioRender.com”, 22 December 2021.

**Figure 3 cells-11-00429-f003:**
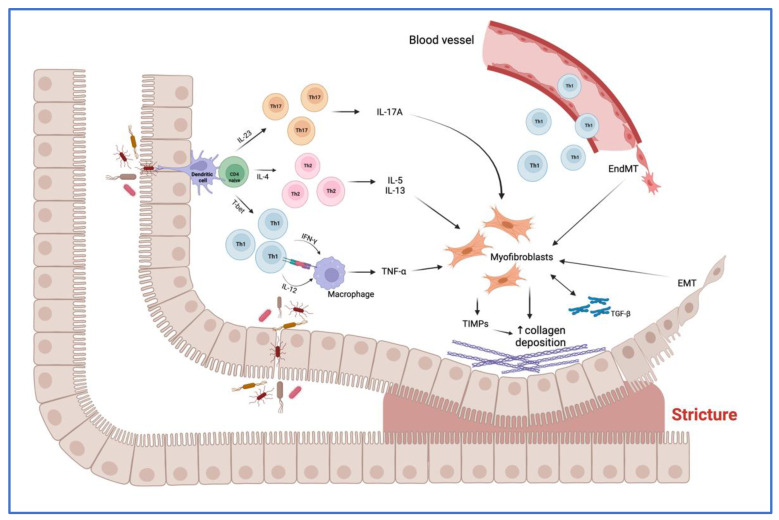
Main molecular and cellular mechanisms, and their interaction, underlying the fibrogenic process leading to stricture formation in Crohn’s disease (CD). Intestinal mucosal infiltration of CD4+ T cells represents a key characteristic of CD. Multiple Th subsets have been identified, with different role in the fibrogenic process. The cross-talk between macrophage and T cells, sustained by Th1 pro-inflammatory cytokines, including interferon (IFN)-ɣ and interleukin (IL)-12, results in the production of tumor necrosis factor (TNF)-α, which promotes myofibroblast production of transforming growth factor (TGF)-β1. The latter inhibits the production of matrix metalloproteinases (MMPs) and favors the production of tissue inhibitor of metalloproteinases (TIMPs), which causes abnormal collagen deposition, with consequent fibrosis and stricture formation. In addition, Th2 and Th17 cells have a pro-fibrotic role through the production of pro-fibrotic cytokines, especially IL-17A, which induces intestinal myofibroblast secretion of collagen and TIMPs and significantly inhibits myofibroblast migration. The fibrotic process is also sustained by epithelial-mesenchymal transition (EMT) and endothelial-mesenchymal transition (EndMT), constantly evolving processes in which epithelial and endothelial cells acquire fibroblast characteristics. Abbreviations: EMT, epithelial-mesenchymal transition; EndMT, endothelial-mesenchymal transition; IL, interleukin; IFN, interferon; T-bet, T-box transcription factor; TGF, transforming growth factor; Th, T helper cell; TIMP, tissue inhibitor of metalloproteinase; TNF, tumor necrosis factor; ↑, increase. Created with “BioRender.com”, 21 January 2022.

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
