# Peer review of "Therapeutic Targeting of Intestinal Fibrosis in Crohn’s Disease"

_cells, 2022, doi:10.3390/cells11030429_

Round 1

Reviewer 1 Report

The present article is a narrative review of the fibrotic phenomenon in crohn's disease, focusing mainly on therapeuthic targets and future perspectives. 
The topic is very much of interest, currently a "hot topic" in the field. The review is very well organized and written The authors touch most apsect of the problem and their overview is weel-balanced and at the same time very detailed. Some minor comments:
-Review methodology could be intergrated with more details on search methods, possibly with a figure, although if the search was not systematic (this should be clearly stated!!) this could also be avoided.
-Pathogenesis part is very well done, bot detailed and fluent. A minor imprecision is the absence of reference to IL-25 (which has also been shown to be an actor in the process) and its intercation with TGF-B and TNF-a (Fina D, Interleukin-25 production is differently regulated by TNF-α and TGF-β1 in the human gut. Mucosal Immunol. 2011 ).
-A concise section on diagnosing fibrosis might be appropriate. A main problem in current CD management is the distinction between inflamamtion and fibrosis, which today is still very difficult. This has been briefly cited in the major challenges section but may deserve some more attention.Furthermore, In the postoperative setting fibrosis recurrence may be even more difficult to daignose and this should also be noted as most patients undergo surgery - no single method exists today to diagnose fibrotic recurrence adn in particualr its contibution, pros and cons of endoscopy and imaging could be briefly cited (Onali S, Endoscopic vs ultrasonographic findings related to Crohn's disease recurrence: a prospective longitudinal study at 3 years. J Crohns Colitis. 2010).
-The therapeuthic section, although incentrated on new targets should give a little more space to existing strategies. Endoscopic and surgical solutions should be briefly explained. Many surgical solutions are being explored to reduce fibrosis recurrence including special anastomotic configurations(Katsuno H, Novel antimesenteric functional end-to-end handsewn (Kono-S) anastomoses for Crohn's disease: a report of surgical procedure and short-term outcomes. Dig Surg. 2015), mesenteric/lymph-nodal excision (Coffey CJ, Inclusion of the Mesentery in Ileocolic Resection for Crohn's Disease is Associated With Reduced Surgical Recurrence. J Crohns Colitis. 2018 N.) and the use of laparoscopy which may decrease or at least slow down the process (Sica GS, Laparoscopic versus open ileo-colonic resection in Crohn's disease: short- and long-term results from a prospective longitudinal study. J Gastrointest Surg. 2008).
The rest of the discussion on therapeutic targeting is brilliantly exposed
-Major challenges: endpoint section could desrve a little more explanation.
-It would be interesting to add a table with ongoing phase III clinical trials, if any can be found.
-Figures and tables are ok

Author Response

The present article is a narrative review of the fibrotic phenomenon in Crohn's disease, focusing mainly on therapeutic targets and future perspectives. The topic is very much of interest, currently a "hot topic" in the field. The review is very well organized and written The authors touch most aspect of the problem and their overview is well-balanced and at the same time very detailed.

We thank the reviewer for the comments and recommendations. We have carefully addressed the issues raised as follows:

Some minor comments:

-Review methodology could be integrated with more details on search methods, possibly with a figure, although if the search was not systematic (this should be clearly stated!!) this could also be avoided.

We thank the Reviewer for the comment. This is not a systematic review but rather an expert-based, narrative one and now this is clearly stated. We have expanded the section of material and methods, better clarifying our search strategy. For completeness, we have now added the number of papers evaluated in the present study and we have provided a diagram showing the search strategy.

-Pathogenesis part is very well done, both detailed and fluent. A minor imprecision is the absence of reference to IL-25 (which has also been shown to be an actor in the process) and its interaction with TGF-B and TNF-a (Fina D, Interleukin-25 production is differently regulated by TNF-α and TGF-β1 in the human gut. Mucosal Immunol. 2011 ).

We thank the Reviewer for his/her suggestion. As requested, we have now added to the section 3.2 details regarding the role of IL-25 in the CD fibrogenesis process.    

-A concise section on diagnosing fibrosis might be appropriate. A main problem in current CD management is the distinction between inflammation and fibrosis, which today is still very difficult. This has been briefly cited in the major challenges section but may deserve some more attention. Furthermore, In the postoperative setting fibrosis recurrence may be even more difficult to diagnose and this should also be noted as most patients undergo surgery - no single method exists today to diagnose fibrotic recurrence and in particular its contribution, pros and cons of endoscopy and imaging could be briefly cited (Onali S, Endoscopic vs ultrasonographic findings related to Crohn's disease recurrence: a prospective longitudinal study at 3 years. J Crohns Colitis. 2010).

We appreciate the Reviewer comment and we have now included a new section on the diagnosis of intestinal fibrosis (section 4) with reference also to the postoperative setting as suggested.

-The therapeutic section, although incentrated on new targets should give a little more space to existing strategies. Endoscopic and surgical solutions should be briefly explained. Many surgical solutions are being explored to reduce fibrosis recurrence including special anastomotic configurations(Katsuno H, Novel antimesenteric functional end-to-end handsewn (Kono-S) anastomoses for Crohn's disease: a report of surgical procedure and short-term outcomes. Dig Surg. 2015), mesenteric/lymph-nodal excision (Coffey CJ, Inclusion of the Mesentery in Ileocolic Resection for Crohn's Disease is Associated With Reduced Surgical Recurrence. J Crohns Colitis. 2018 N.) and the use of laparoscopy which may decrease or at least slow down the process (Sica GS, Laparoscopic versus open ileo-colonic resection in Crohn's disease: short- and long-term results from a prospective longitudinal study. J Gastrointest Surg. 2008).
The rest of the discussion on therapeutic targeting is brilliantly exposed

We appreciate the Reviewer comment and we have implemented the 4.2 section focusing on endoscopic and surgical management as suggested.

-Major challenges: endpoint section could deserve a little more explanation.

We thank the Reviewer for his/her suggestion. The endpoint section of chapter 5 has now been expanded.

-It would be interesting to add a table with ongoing phase III clinical trials, if any can be found.

We thank the Reviewer for his/her interesting suggestion. Despite the urgency of anti-fibrotic therapy and the numerous molecules identified as potential anti-fibrotics, when we searched on “clinicaltrials.org” using the terms "Crohn’s disease", "intestinal fibrosis" and "anti-fibrotic therapy" no ongoing or recruiting trials were found. This has now been specified in section 5.  

-Figures and tables are ok”

Thank you for the thorough correction.

Reviewer 2 Report

Paper describes relevant clinical problem in CD, is well written and organised. I have one remark. In my opinion paper would benefit from the more extensive presentation of the search strategy. I am aware that it is not systematic review, however, I think that Authors could provide diagram demonstrating applied method.

Author Response

“Paper describes relevant clinical problem in CD, is well written and organised.I have one remark. In my opinion paper would benefit from the more extensive presentation of the search strategy. I am aware that it is not systematic review, however, I think that Authors could provide diagram demonstrating applied method.”

We thank the Reviewer for the comments and recommendations. This is not a systematic review but rather an expert based, narrative one and now this is clearly stated. We have expanded the section of material and methods, better clarifying our search strategy. For completeness, we have now added the number of papers evaluated in the present study and we have provided a diagram showing our search strategy.

Reviewer 3 Report

I read this review with interest. This is challenging but exciting and interesting theme.  

I think the contents covered whole area regarding intestinal fibrosis and were well written.

However, each factor was not  linked each other. Some of them should be linked. I recommend adding the Figure which shows the relation between fibrosis and factors with listed factors.

Cytokine and antibody of issues were well written. Please describe the role of immunocyte like Th cells and Treg with IL-17 for fibrosis in more detail.

Author Response

“I read this review with interest. This is challenging but exciting and interesting theme. I think the contents covered whole area regarding intestinal fibrosis and were well written.

We thank the Reviewer for his/her interest and attention in correcting our review. We have addressed his concerns as follows: 

However, each factor was not  linked each other. Some of them should be linked. I recommend adding the Figure which shows the relation between fibrosis and factors with listed factors.

We thank the Reviewer for his/her interesting suggestion. Although many mechanisms underlying intestinal fibrosis are known, the correlation between them is in most cases unknown and elusive, and now this is clearly stated. As suggested, a new figure has been added (Figure 2) that shows some of the known cellular and molecular mechanisms and their correlation in the fibrogenic process leading to intestinal stricture formation. 

Cytokine and antibody of issues were well written. Please describe the role of immunocyte like Th cells and Treg with IL-17 for fibrosis in more detail.”

We thank the Reviewer for his/her comment. As requested, we have now added to the section 3.2 details regarding the role of Th cells and T-reg, focusing on the relationship with IL-17 in intestinal fibrosis.

Round 2

Reviewer 3 Report

Authors modified their manuscript well. I do not have any additional comment.